# Simulating Weak Attacks in a New Duplication–Divergence Model with Node Loss [note 1]

**DOI:** 10.3390/e26100813

**Published:** 2024-09-25

**Authors:** Ruihua Zhang, Gesine Reinert

**Affiliations:** 1Department of Statistics, University of Oxford, 24–29 St. Giles’, Oxford OX1 3LB, UK; 2The Alan Turing Institute, London NW1 2DB, UK

**Keywords:** duplication–divergence model, gene loss, weak attack, protein–protein interaction networks

## Abstract

A better understanding of protein–protein interaction (PPI) networks representing physical interactions between proteins could be beneficial for evolutionary insights as well as for practical applications such as drug development. As a statistical model for PPI networks, duplication–divergence models have been proposed, but they suffer from resulting in either very sparse networks in which most of the proteins are isolated, or in networks which are much denser than what is usually observed, having almost no isolated proteins. Moreover, in real networks, where a gene codes a protein, gene loss may occur. The loss of nodes has not been captured in duplication–divergence models to date. Here, we introduce a new duplication–divergence model which includes node loss. This mechanism results in networks in which the proportion of isolated proteins can take on values which are strictly between 0 and 1. To understand this new model, we apply strong and weak attacks to networks from duplication–divergence models with and without node loss, and compare the results to those obtained when carrying out similar attacks on two real PPI networks of *E. coli* and of *S. cerevisiae*. We find that the new model more closely reflects the damage caused by strong and weak attacks found in the PPI networks.

## 1. Introduction

From virtual internet to practical traffic control systems, from small social networks to large biological systems, networks are ubiquitous, and so are attacks on networks. For example, an internet cyber attack can slow down information transmission or cause information leakage, and drugs can target a number of different proteins. Reference [1] shows that partial inactivation of multiple nodes simultaneously in a network can be more effective than the complete elimination of a node, by measuring the sum of the inverse of the shortest path between any two nodes of biological networks (the *network efficiency*).

This result motivates the study of weak attacks in pharmaceutical designs. For example, broader-specificity, lower-affinity compounds or multidrug therapies may cause larger damage in network efficiency than high-affinity, high-specificity compounds. The success of multitarget drugs, like non-steroidal anti-inflammatory drugs (NSAIDs) [2], metformin [3], and Gleevec [4], to treat diseases including AIDS, cancer, atherosclerosis, and Alzheimer’s disease, all suggest that attacking multiple targets may be a useful therapeutic strategy.

To anticipate the effect of an attack, a well-fitting parametric network model could help gain insights. For protein–protein interaction (PPI) networks, duplication–divergence (DD) models have been suggested, see for example [5,6,7]. This paper hence starts with practically simulating weak attacks in a duplication–divergence model. Simulations from [8] suggest that DD models can generate networks which resemble PPI networks more than a basic Bernoulli random graph model. However, ref. [9] found that while Monte Carlo tests based on network comparison statistics do not reject the DD model for some small-virus PPI networks, they do reject it (at the 5% level) for *E. coli*, *worm*, *fly*, *S. cerevisiae*, and *human* PPI networks. Indeed, DD models are known not to be very realistic; for example, ref. [10] proved that as the number of nodes tends to infinity, the proportion of isolated nodes in a standard DD model converges to either 0 or 1, neither of which is realistic.

To understand theoretically how weak attacks damage PPI networks, it is instructive to consider a simple Bernoulli G(n,M) random graph with *n* nodes and *M* edges. We derive a Poisson approximation for the number of isolated nodes in a G(n,M) via Stein’s method, which gives explicit bounds in total variation distance, and we prove similar bounds for the number of isolated nodes after different attack strategies. These results lead to a clear statistical rejection of the hypothesis that the real PPI networks in this paper follow a G(n,M) model.

To identify a more realistic model for PPI networks, we notice that the current DD models ignore gene losses, a biological function [11] which can potentially balance the proportion of isolated nodes. As genes code for proteins, it is plausible that a model with node loss may perform better than standard duplication–divergence models for PPI networks. This paper introduces a new DD model with node loss, where a node can be lost with probability *q* if it is isolated. We compare the simulation results of weak attacks in a standard DD model and the DD model with node loss, and conclude that the new model indeed generates a more realistic performance.

This paper is structured as follows. Section 2 describes the datasets and attack strategies that are employed, as well as the damage strategy and measures of damage. Section 4 introduces the new DD model with node loss. Simulations of various attack strategies on PPI networks on real and synthetic networks are provided in Section 5. The results are discussed in Section 6. Appendix A contains details of the Poisson approximation results and Appendix B contains additional figures. The code is available at https://github.com/rh-zhang/Entropy_CNC2023 (accessed on 24 August 2024).

## 2. Data and Methods

### 2.1. Datasets

We use PPI networks for *E. coli* and *S. cerevisiae* downloaded from STRING (version 12.0, accessed on 11 March 2024), restricted to physical interactions between proteins only. The resulting networks are unweighted, undirected physical subnetworks representing direct interactions between proteins only, excluding indirect functional associations. We remove interactions with a STRING score [12] less than 0.500 for the *E. coli* PPI network and less than 0.400 for the *S. cerevisiae* PPI network, taking all evidence channels into account. The 0.400 threshold is the default threshold in STRING; the 0.500 threshold for *E. coli* is chosen such that the number of isolated nodes is of a similar magnitude (around 1100) in both networks, see Table 1. As shown in Figure 1, the number of isolated nodes increases as the threshold of STRING scores increases. However, the overall trend regarding the impact of weak attacks on the networks remains consistent in our results, as shown in Figure A9.

We note that there is no claim that all possible protein–protein interactions have been detected, and hence the STRING database is unlikely to contain all true interactions; it may also contain some false positive interactions. Our study is conceptual and hence not severely affected by such false positives and false negatives, under the assumption that there is no strong systematic connection between errors in the data and isolated proteins.

We assign a uniform weight of 1 to all the remaining edges in the datasets, with the summary statistics shown in Table 1. The reason for ignoring weights is conceptual simplicity.

### 2.2. Attack Strategies

The attack strategies used in this paper follow those from [1]. While in [1], networks with weighted edges are allowed, in our investigative study we set all edge weights equal to 1 initially; some attacks lead to a reduction in some of the edge weights. The attack strategies are split into three categories.

**Type A:** Complete knockout: the attack of a single target by eliminating all interactions of a given node, as shown in Figure 2A.

**Type B:** Partial inactivation of a target, as shown in Figure 2: B, which is modelled in two different ways:

B1: Partial knockout: half of the interactions of a given node are removed (the number of interactions removed is rounded down when the degree of the target is odd). If a node is attacked partially once, it will not be attacked again to ensure no node is completely knocked out. This is shown in Figure 2B1.

B2: Attenuation: all interactions of a given node are attenuated by halving their weight.

**Type C:** A distributed, system-wide attack, which can affect any interactions (i.e., edges) within a network. Again, such an attack is modelled in two different ways:

C1: Distributed knockout: edges are deleted independently at random, with the same deletion probability, as shown in Figure 2C1.

C2: Distributed attenuation: edges are chosen independently at random, with the same probability, and their weights are halved.

These attacks can be interpreted in pharmaceutical terms; a high-affinity drug completely eliminates an interaction while a low-affinity drug attenuates it, and a highly specific drug targets one single interaction only, while less specific drugs affect some or all interactions of a given node.

### 2.3. Successive Maximal Damage Strategy

As in [1], the nodes being attacked in the simulation of this paper are selected based on a successive maximal damage strategy. The search for maximal damage caused by multiple attacks is computationally very intensive. For instance, to determine which 5 of the 1000 edges of a given network need to be deleted in order to produce a maximal effect on the network efficiency, one would need to test $1000!/\left(5!995!\right)\approx 8.25\;\times \;10^{12}$ cases in a single-simulation experiment.

Instead, we use a greedy algorithm: for each type of attack, in each step we choose the action that produces the largest damage. The greedy algorithm is carried out by first determining the damage caused by the removal of each individual node or edge, depending on the strategy. The node or edge causing the maximum damage is selected for removal in the subsequent attack. We note that the damage calculated in this manner is only an estimate of the maximal damage, since there may be more efficient combinations.

### 2.4. Measures of Damage

The damage induced by the attacks on the networks is measured by three metrics: the network efficiency for the transcriptions regulator networks, as used in [1], the average number of edges in the 1-step ego network for the PPI networks, as proposed in [13] for assessing the robustness of network metrics, and the number of isolated nodes.

The **network efficiency** (NE) of an undirected, unweighted graph of *n* nodes is $\sum_{i\neq j}\frac{1}{d_{ij}}$, where $d_{ij}$ is the length of a shortest path between nodes *i* and *j*. If the network is weighted, $d_{ij}$ is the weight of a path between nodes *i* and *j* with a minimum weight. If any two nodes i≠j are disconnected, then $d_{ij}=\infty$, and their contribution to the calculation of network efficiency is 0. NE measures how efficiently a network exchanges information. The underlying idea is that the more distant two nodes are in a network, the less efficient their exchange of information will be.

The second measure is the **average number of edges in the 1-step ego network,** where a 1-step ego network consists of a focal node (the *ego*), the nodes to which the ego is directly connected (the *alters*), and the edges, if any, among the alters.

The third measure is the **number of isolated nodes.** We add this measure because an ideal attack would isolate a deleterious node. Moreover, in a Bernoulli random graph model this measure can be analysed analytically and thus is useful for providing theoretical underpinning.

### 2.5. Bernoulli Random Graphs

Given the number *n* nodes and the number *M* of edges in a simple network, in the absence of further information one may model the network as a G(n,M) graph. This is a random graph that is chosen uniformly at random from the collection of all simple graphs which have *n* nodes and *M* edges, where 0≤M≤n2.

The distribution of the degree of a node *v*, D(v), in a G(n,M) graph is hypergeometric; there are n−1 edges that are adjacent to *v*, out of the n2 potential edges, of which we choose *M*. Abbreviating the number of node pairs by N=n2 we thus have
P(D(v)=k)=n−1kN−(n−1)M−kNM,k=0,1,…,M.

We can calculate the expected number of isolated nodes from this distribution, but not its variance, due to the dependence between edges. To clarify the dependence, for example, if we know that the first n−1 nodes have degree 0, then node *n* necessarily must have degree 0. As this dependence is usually weak, we derive a Poisson approximation for the number of isolated nodes in the total variation distance. The *total variation distance $d_{TV}$* measures the largest absolute difference between the probabilities of the actual probability distribution and the Poisson approximation. For distributions *P* and *Q* taking values in $Z_{+}=\left\{0,1,\ldots \right\},$ the total variation distance is defined as
(1)$$d_{TV}\left(P,Q\right)=\underset{A\subset Z_{+}}{\sup}\left|P\left(A\right)-Q\left(A\right)\right|.$$
For M≤N−(n−1), the probability that node *i* is isolated is
P(Ii=1)=N−(n−1)MNM:=π.
With *W* denoting the number of isolated nodes, its expectation is E(W)=nπ=:λ, and this is the parameter which we choose for the approximating Poisson distribution.

**Theorem 1.** 
*It holds that*

dTV(L(W);Po(λ))≤min(1,λ−1)e−np(1+n−2N+2−n)+p(1+n−2N+2−n)+2−3n+n2N+2−n=1+np1+n+Np−2N−Np−n+2.



This bound tends to zero as p:=M/N→1. The proof and more details can be found in Section A.1. Section A.1 also gives Poisson approximations for the number of isolated nodes after an attack for the different attack strategies. These results may be of independent interest.

## 3. Duplication–Divergence Models

Simulations suggest that duplication–divergence (DD) models generate networks which provide a better fit to protein interaction networks than the standard models [8]. There are different variations of duplication–divergence models in the literature, see for example [6,7,14,15]. Here, we use a version, from [15], which incorporates the parameters of the probability of edge divergence, *p*, but we exclude the possibility of a parent–child edge.

A standard duplication–divergence model $DD(t_{0};p)$ starts from a complete graph $G_{t_{0}}$ on $t_{0}$ nodes (labelled from 1 to $t_{0}$), and then repeats the following steps until a graph of the desired size is obtained:**Duplication:** at time *t*, a node *u* is selected uniformly at random. A node labelled as t+1 is added, as well as the edges between node t+1 and the neighbours of node *u* in the graph.**Divergence:** edges involving node t+1 are randomly retained with probability *p*.

An illustration of a DD model is shown in Figure 3.

Reference [15] found that the degree distributions of the DD model described above are in reasonable agreement with the distributions observed in real protein networks, and tuning the parameter *p* reveals a rich behaviour of the model. When *p* is large, the network growth lacks self-averaging and results in a great diversity of networks grown out of the same initial condition. For p<0.5, the average degree increases very slowly or tends to a constant, and the degree distribution has a power-law tail. Several real protein–protein networks are estimated to have a *p* value of around 0.4 [15]. As shown in Figure A1, the choice of *p* does not affect the qualitative behaviour of the models against attacks.

## 4. A New Duplication–Divergence Model Which Allows for Node Loss

Although simulations have shown that the DD model described above is more realistic than a G(n,M) model, ref. [10] proved that the proportion of isolated nodes in a DD model either converges to 0 or 1. This behaviour does not match biological intuition, and other network models do not exhibit it; for example, we prove in Appendix A that the proportion of isolated nodes in a G(n,M) model does not have to converge to either 0 or 1.

The quality of a network model has to be judged by the research question to be addressed. In a series of Monte Carlo tests for *E. coli*, *worm*, *fly*, *S. cerevisiae*, and *human* PPI networks and some small-virus PPI networks [9], a DD model (allowing for a non-zero probability of a parent–child edge) is rejected as a model for the large PPI networks based on network comparison statistics including graphlet correlation distance, graphlet degree distribution agreement, Netal, and Netdis. In contrast, in the small-virus PPI networks investigated in [9], the DD model is not rejected by most of these network comparison statistics. These statistics do not include the number of isolated nodes, but Netdis is based on subgraph counts in ego networks, and is thus related to our outcome measure of the average number of edges in 1-step ego networks. Hence, these Monte Carlo results indicate that the DD model may not be a good fit for larger PPI networks when the interest is in modelling the effect of attacks.

From a biological viewpoint, genes and the proteins they code for can not only duplicate, but can also be lost. For example, gene loss can occur during natural mutations and frameshifts [16]. Furthermore, many examples support the idea that gene loss can be an adaptive evolutionary force that is especially common when organisms are faced with abrupt environmental challenges [11]. Adaptive gene loss, or gene loss in general, can be of potential interest in the study of both biomedicine and evolution.

Therefore, we modify the DD model to allow for both node addition and for the loss of nodes. In addition to the process that generates a DD model, a node loss step is added after every duplication-and-divergence step. In particular, we focus on the node loss mechanism that a node can be lost with probability *q* if it is isolated.

**Duplication:** at time *t*, a node *u* is selected uniformly at random. A node labelled as t+1 is added, as well as the edges between node t+1 and the neighbours of node *u* in the graph.**Divergence:** edges involving node t+1 are randomly retained with probability *p*.**Node loss:** a node is randomly lost with probability *q* if it is isolated.

A graph illustration of our new model is present in Figure 4.

## 5. Results

### 5.1. Simulation of Weak Attacks in Real PPI Networks

Here, we apply the various attack strategies to our PPI networks datasets with 10 repeats. Figure 5 shows that as for the PPI networks of *E. coli* and *S. cerevisiae* the number of targets that are subject to weak attacks increases, and the damage caused by weak attacks becomes larger and is significantly greater than the damage caused by complete knockout.

To understand the expected effects of attacks, a parametric model may be useful. Next, we investigate two such models.

### 5.2. The Number of Isolated Nodes in a Bernoulli Random Graph

As a baseline model for a PPI network, we use a G(n,M) model. In Appendix A we derive an upper bound for the total variation distance for the number of isolated nodes in real PPI networks using a G(n,M) graph under Poisson approximation, see Section A.1. The Poisson approximation comes with an explicit bound, which we abbreviate here as Δ, on the total variation distance (Equation 1). If *W* denotes the number of isolated nodes, λ its expectation under the G(n,M) model, and *Z* a Poisson-distributed random variable with mean λ, then it follows that for all *k*,
P(Z≥k)−Δ≤P(W≥k)=P(Z≥k)+(P(W≥k)−P(Z≥k))≤P(Z≥k)+Δ.

Thus, the Poisson approximation can be used to assess statistical significance.

For our *E. coli* and *S. cerevisiae* data, the estimated upper bound for the total variation distance is $3.73\times 10^{-15}$ and $9.28\times 10^{-19}$, respectively. While these bounds are small, the *p*-values associated with these bounds are 0 up to 6 significant digits under a two-sided test in which the null hypothesis of the G(n,M) model is rejected for very small or very large numbers of isolated nodes, lending evidence to the explanation that the G(n,M) model does not explain the observed number of isolated nodes well. The observed number of isolated nodes in *E. coli* and *S. cerevisiae* is 833 and 1100, respectively, whereas the expected number of isolated nodes under the G(n,M) model is $2.99\times 10^{-14}$ and $1.27\times 10^{-17}$. This suggests that a G(n,M) graph may not be suitable for modelling these real PPI networks when the interest is in the number of isolated nodes as a summary statistics.

We further derived upper bounds for the total variation distance under Poisson approximation for the number of isolated nodes after different types of attack, see Section A.2, Section A.3 and Section A.4. Again, the results are highly significant, with *p*-values equal to 0 up to 6 significant digits, indicating that after an attack, the G(n,M) model still does not fit the data well. Hence, a different model for the data is needed. Next, we investigate the standard duplication–divergence model from Section 3.

### 5.3. Simulation of Weak Attacks in Duplication–Divergence Model

In this section, we present the simulation results of applying weak attacks to realisations of the standard duplication–divergence model $DD(t_{0};p)$ from Section 3. The model is undirected, and all edges are set to have unit weight; we take $t_{0}=3$, and start the simulation of the graph with a triangle. This choice ensures that the generated networks can include triangles, resulting in non-zero local and global clustering coefficients; thus they are able to match this key characteristic of PPI networks. In contrast, if the graph is initiated with just a connected pair of nodes, the generated graphs cannot have any triangles; the corresponding simulation results, shown in Appendix B, are, however, similar regarding the effect of attacks. Reference [10] proves that $p^{*}$ solving the equation $pe^{p}=1$ is a critical value, in the sense that for $p>p^{*}$ there is no limiting degree distribution. In this paper, we take *p* to be 0.4, a value smaller than $p^{*}\approx 0.567$. The simulations are run for 1000 steps, with five repeats.

The top two plots of Figure 6 show how partial attacks damage a DD network compared to complete knockout attacks. As illustrated in the top left plot of Figure 6, while increasing the number of nodes being attacked weakly eventually enhances the damage efficiency for a large number of attacks, complete knockout attacks serve as a robust method to destroy the network.

The bottom two plots of Figure 6 show how distributed attacks damage a DD network compared to complete knockout attacks. The horizontal line representing the damage caused by one complete knockout suggests that the effect of 6 distributed knockout or 13 distributed attenuation attacks is equivalent to the effect of one complete knockout. This indicates that distributed attacks are less effective than both complete knockout attacks and partial attacks.

### 5.4. Simulation of Weak Attacks in the New Node Loss Model

Now, we present the simulation results of applying weak attacks onto the new node loss model introduced in Section 4. Again, the model is undirected with all edges assigned unit weight. The simulations are run with 10 repeats and the average network efficiency values are reported to account for randomness. We note here that we did not carry out a grid search for the optimal parameter choices for the DD models without and with gene loss for the different organisms, as the focus of this paper is the qualitative behaviour of the new DD model with gene loss, and not detailed modelling of observed PPI networks.

Figure 7 shows the results for p=0.4 and q=0.2 under different weak attacks. We observe that in 25 attacks, a complete knockout attack is more effective than a partial attenuation when half of the edges connected to one node are eliminated, but less effective than a partial attenuation when halving two nodes or five nodes. Our results indicate that as the number of halved nodes increases, the weak attacks damage networks more efficiently. Furthermore, distributed attacks are less effective than complete knockout and partial attacks, mirroring the qualitative impact observed in real PPI networks.

We observe that the pattern of Figure 7 for the new node loss model is more similar to the pattern of Figure 5 for the real datasets than the pattern of Figure 6 for a standard DD model. This suggests that the new node loss model can mimic the effect of weak attacks on protein–protein interaction networks more realistically than the standard $DD(t_{0},p)$ model.

Regarding the effect of the probability of node loss on weak attacks in the new node loss model, we notice that the number of distributed attacks required to achieve the equivalent effect as one complete knockout attack increases as *q* increases. This raises a natural question regarding how the value of *q* affects the efficiency of weak attacks in the new node loss model. In our simulations, shown in Figure 8, the resilience of the new node loss model to weak attacks results in a slower rate of network degradation. This can be attributed to the fact that higher *q* values correspond to an increased likelihood of losing isolated nodes, which in turn leads to a more connected graph structure.

## 6. Discussion

In this paper, we have assessed standard models for PPI networks and we have introduced a new node loss model which is motivated by observed gene loss in organisms. We show that our new node loss model captures the effect of weak attacks in a protein–protein interaction network more realistically than a standard DD model (i.e., *q* = 0).

To further enhance the robustness of our results, as future work we aim to derive analytical results for the average number of edges in a 1-step ego network and for the network efficiency before and after attacks in the new node loss model.

It is perhaps not surprising that the new node loss model performs better due to its incorporation of a natural and common biological adaptation, namely, gene loss, occurring throughout evolution. As a next step, variants of the new node loss model could be examined; for example, one could include the case where the probability of a parent–child node edge is not zero. In order to understand how node loss affects duplication–divergence behaviour, we also aim to investigate other parameters that can affect a node loss in a network; for example, a pair of nodes may be more likely to be lost if they are connected by an isolated edge.

Regarding the network representation of PPIs, we chose the PPI networks from the STRING database, which represents each protein-coding gene locus by only a single, representative protein. The datasets contain non-binary data which could be incorporated in the analysis. Moreover, future work will assess the effect of restricting the protein interactions from the STRING database to physical interactions, by repeating the analysis for the full STRING PPI networks. Hypergraph representations as in [17] may also be fruitful. 

## Figures and Tables

**Figure 1 entropy-26-00813-f001:**
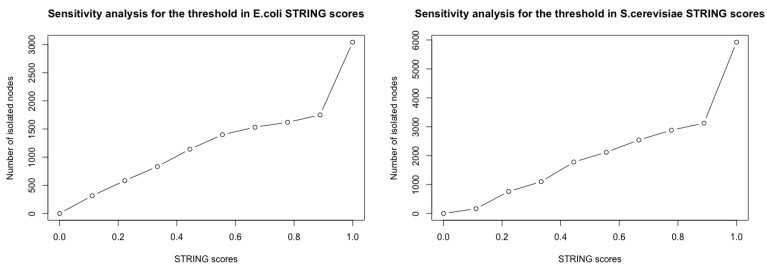
Sensitivity analysis for the number of isolated nodes in the *E. coli* and *S. cerevisiae* PPI networks across varying STRING score thresholds.

**Figure 2 entropy-26-00813-f002:**
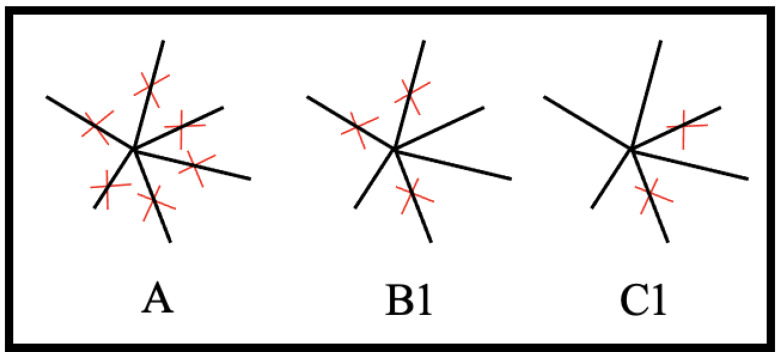
Attack strategies. (**A**) Complete knockout attack: all edges connected to the attacked node are eliminated. (**B1**) Partial knockout attack: half of the edges connected to the attacked node are eliminated. (**C1**) Distributed knockout attack: randomly selected edges are eliminated. Adapted from FIG.1 in [1].

**Figure 3 entropy-26-00813-f003:**
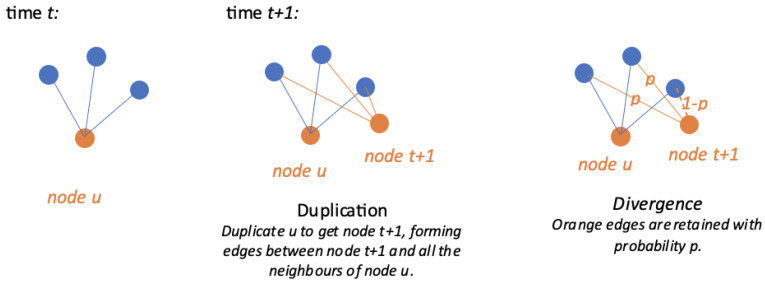
Graph illustration of a duplication–divergence model.

**Figure 4 entropy-26-00813-f004:**
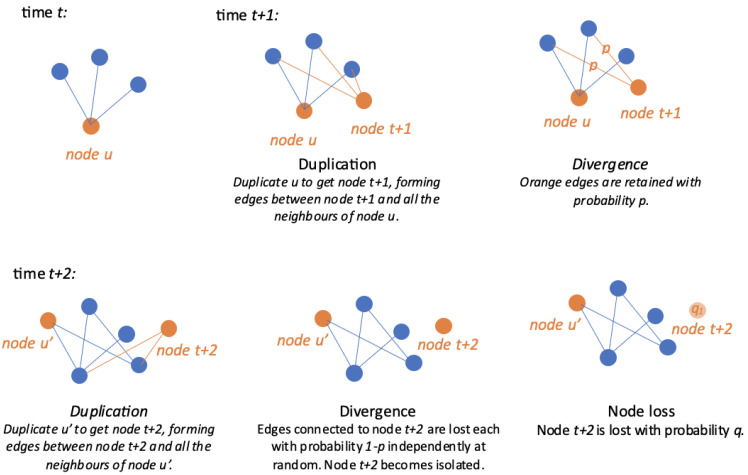
Graph illustration of a new duplication divergence model with node loss.

**Figure 5 entropy-26-00813-f005:**
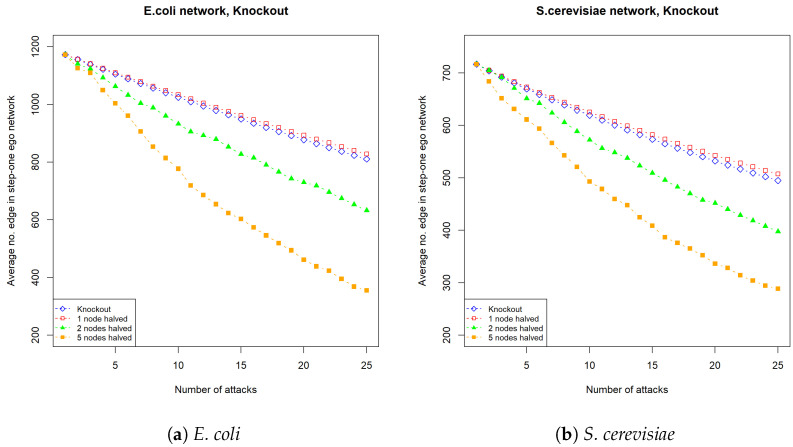
The average number of edges in the 1-step ego network of an *E. coli* and *S. cerevisiea* PPI network after 25 attacks. (**a**) shows the average number of edges in the 1-step ego network in a *E. coli* PPI network under 25 knockout attacks. Blue line: complete knockout; red line: partial knockout with half of the edges connected to one node being removed at each attack; green line: partial knockout with half of the edges connected to two nodes being removed at each attack; orange line: partial knockout with half of the edges connected to five nodes being removed at each attack. (**b**) shows the average number of edges in the 1-step ego network in a *S. cerevisiae* PPI network under 25 attenuation attacks. Since a one-node halved knockout only deletes half of the edges connected to the selected node, when a node has a degree of at least 2 it causes less damage than a complete knockout which removes all the edges connected to the selected node.

**Figure 6 entropy-26-00813-f006:**
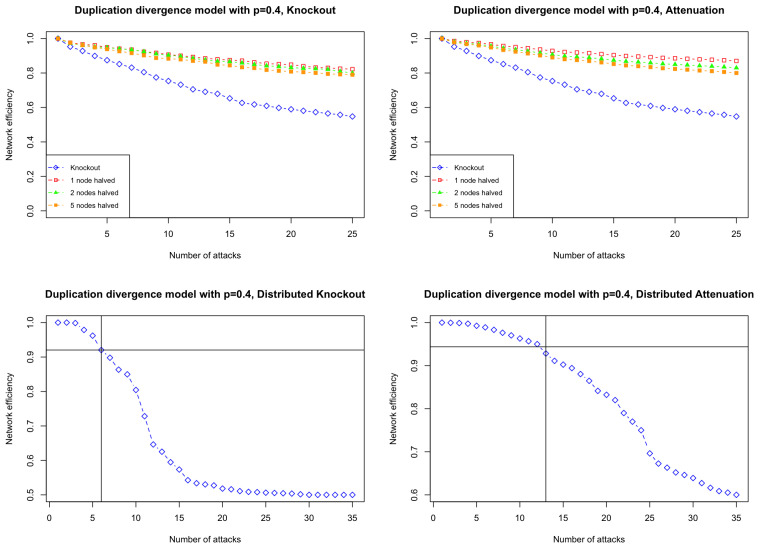
Network efficiency after up to 25 weak attacks on simulations from the duplication–divergence model starting with a triangle with a divergence rate p=0.4. **Top left:** knockout attacks. Blue line: complete knockout; red line: partial knockout with half of the edges connected to one node being removed at each attack; green line: partial knockout with half of the edges connected to two nodes being removed at each attack; orange line: partial knockout with half of the edges connected to five nodes being removed at each attack. **Top right:** attenuation attacks. Blue line: complete knockout; red line: partial attenuation with all the edges connected to one node being halved at each attack; green line: partial attenuation with all the edges connected to two nodes being halved at each attack; orange line: partial attenuation with all the edges connected to five nodes being halved at each attack. **Bottom left**: distributed attacks, with edges drawn from a random distribution; the horizontal line represents equivalent damage to the network achieved by one complete knockout. **Bottom right**: distributed attenuation attacks, with the weight of edges drawn from a random distribution to be halved; the horizontal line represents equivalent damage to the network achieved by one complete knockout.

**Figure 7 entropy-26-00813-f007:**
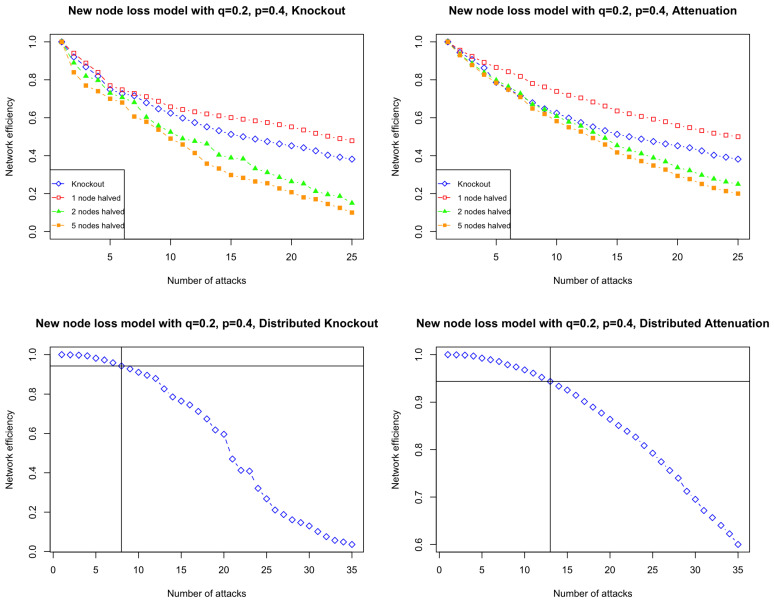
Network efficiency after up to 25 weak attacks on simulations from the new node loss model starting with a triangle; a node can be lost with probability q=0.2, using a divergence rate *p* = 0.4. The graph is undirected and has unit edge weight. **Top left:** knockout attacks. Blue line: complete knockout; red line: partial knockout with half of the edges connected to one node being removed at each attack; green line: partial knockout with half of the edges connected to two nodes being removed at each attack; orange line: partial knockout with half of the edges connected to five nodes being removed at each attack. **Top right:** attenuation attacks. Blue line: complete knockout; red line: partial attenuation with all the edges connected to one node being halved at each attack; green line: partial attenuation with all the edges connected to two nodes being halved at each attack; orange line: partial attenuation with all the edges connected to five nodes being halved at each attack. **Bottom left**: distributed attacks, with edges drawn from a random distribution; the horizontal line represents equivalent damage to the network achieved by one complete knockout. **Bottom right**: distributed attenuation attacks, with the weight of edges drawn from a random distribution to be halved; the horizontal line represents equivalent damage to the network achieved by one complete knockout.

**Figure 8 entropy-26-00813-f008:**
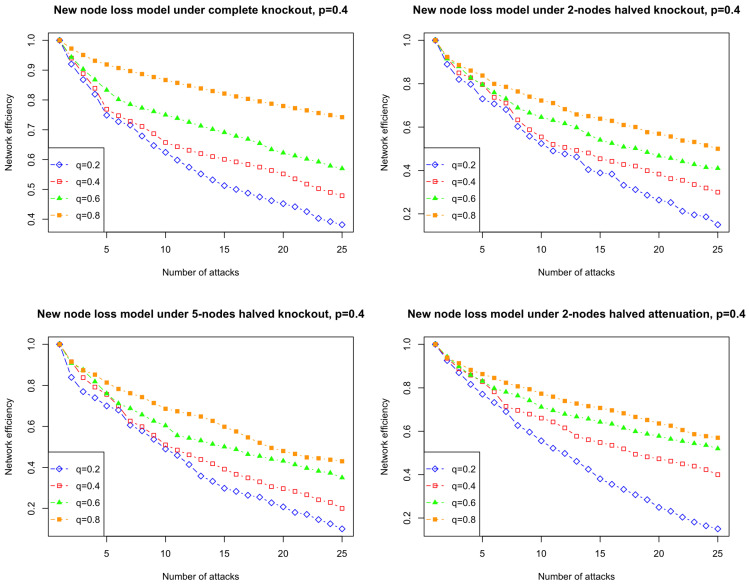
Effect of *q* on the efficiency of weak attacks on simulated networks from the node loss model starting from a triangle with *p* = 0.4, and *q* ranges from 0.2, 0.4, 0.6, to 0.8.

**Table 1 entropy-26-00813-t001:** Summary statistics for the analysed networks; No. stands for *Number of*.

Networks	*E. coli*	*S. cerevisiae*
No. nodes	3043	5925
No. edges	52,914	140,402
No. isolated nodes	1141	1100
Average degree	28.05	59.51
Average local clustering coefficient	0.31	0.40
Global clustering coefficient	0.25	0.80

## Data Availability

The data presented in this study are openly available in GitHub at https://github.com/rh-zhang/Entropy_CNC2023.

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
