# Peer review of "Simulating Weak Attacks in a New Duplication–Divergence Model with Node Loss†"

_entropy, 2024, doi:10.3390/e26100813_

Round 1

Reviewer 1 Report

Comments and Suggestions for Authors

The paper presents a novel duplication divergence model for protein interaction networks and considers the effect of attacks on these networks, showing that they are potentially more representative of the real network response to attacks than traditional DD models. There is also a derivation of the proportion of isolated nodes in a Bernoulli random graph, which shows that they do not model real PPI networks well. However there are some aspects of the work that could be clarified.

Major points:

  - The paper relies on the number of reported isolated nodes in the data sets being accurate. In protein interaction datasets, often only proteins included in experiments are listed, and it is not always certain that all possible protein pairs have been checked for interactions. To my knowledge the S.cerevisiae genome is (slightly) larger than the 5925 nodes listed in the paper. Could the authors comment on whether the PINs they used in the study are known to include all known proteins in the organisms? Further, it may be that although interactions are not reported in the database, the isolated proteins in the data do have interactions that have not yet been detected. Could the authors comment on how this may affect their conclusions?

- A threshold for the edges reported in the STRING database is chosen. Although this is a common practice, could the authors comment on how sensitive the number of isolated nodes is to this choice?

 - The authors only use the response to attacks to justify the new model. Could the authors comment on whether statistics like the degree distribution or clustering coefficient of the simulated DD with node loss model are closer to those of the real networks?

- Why is the parameter of the DD model set to p=0.2? Although the authors show the new model may be more realistic for this parameter setting, it is not clear that this will be the case for a value of p that potentially more closely reproduces the observed networks.

Minor points:

 - In several of the figures (5,6,7) the colours in the legend do not appear to match the colours in the plots. This makes it difficult to follow the results, although from reading the text it becomes clearer.

- In the abstract it could be worth mentioning that existing DD models tend to an isolated node proportion of 0 or 1

- When referring to referenced papers, typically the first author's name is used, for example rather than "[1] have shown", "Agoston et al. [1] have shown".

Comments on the Quality of English Language

The writing is clear and well structured. It could benefit from some minor proof reading as there are a small number of grammatical errors.

Author Response

Dear Reviewer,

We were pleased to receive your report on our submission  and would like to thank you for your insightful review and comments. Please find attached our response to your reviews.

Reviewer 2 Report

Comments and Suggestions for Authors

The author reports a new duplication-divergence model that includes node loss. They apply both strong and weak attacks to networks derived from duplication-divergence models with and without node loss, and compare the results to those obtained from similar attacks on two real PPI networks, E. coli and S. cerevisiae. They found that the new model better reflects the damage caused by strong and weak attacks observed in the PPI networks.

The 5 halved nodes showed slightly different behavior than the others. The authors should provide a higher number of halved nodes to demonstrate converged behavior with respect to the number of nodes.

The legends in Figures 5 and A2 are not correctly labeled. Please double-check all the figures.

Author Response

(The authors gave the same response as above.)
